# FLEXIBLE RESIDUAL BINARIZATION FOR IMAGE SUPER-RESOLUTION

## ABSTRACT

Binarized image super-resolution (SR) has attracted much research attention due to its potential to drastically reduce parameters and operations. However, most binary SR works binarize network weights directly, which hinders high-frequency information extraction. Furthermore, as a pixel-wise reconstruction task, binarization often results in heavy representation content distortion. To address these issues, we propose a flexible residual binarization (FRB) method for image SR. We first propose a second-order residual binarization (SRB), to counter the information loss caused by binarization. In addition to the primary weight binarization, we also binarize the reconstruction error, which is added as a residual term in the prediction. Furthermore, to narrow the representation content gap between the binarized and full-precision networks, we propose Distillation-guided Binarization Training (DBT). We uniformly align the contents of different bit widths by constructing a normalized attention form. Finally, we generalize our method by applying our FRB to binarize convolution and Transformer-based SR networks, resulting in two binary baselines: FRBC and FRBT. We conduct extensive experiments and comparisons with recent leading binarization methods. Our proposed baselines, FRBC and FRBT, achieve superior performance both quantitatively and visually. The code and model will be released.

## 1 INTRODUCTION

Given a full-precision low-resolution (LR) input, single image super-resolution (SR) aims to obtain its high-resolution (HR) counterpart by reconstructing more details. Essentially, image SR is ill-posed, as there exist multiple HR candidates for the same LR input. To address this problem, deep convolutional neural networks (CNNs) and Transformers have been investigated for high-quality reconstructions (Dong et al., 2014; Kim et al., 2016; Lim et al., 2017; Zhang & Patel, 2018; Zhang et al., 2018b; Liang et al., 2021). However, most of them require extensive computational resources, which are usually not friendly for resource-limited devices. In those cases, neural network compression techniques are eagerly needed to significantly reduce model complexity.

As one of the most promising compression methods, binary neural networks (BNNs), where both weights and activations are binarized (*i.e.*, 1-bit binarization), are usually chosen for model deployment (Martinez et al., 2020; Rastegari et al., 2016). Theoretically, BNN enjoys $32\times$ parameter compression ratio and up to $58\times$ computation operation reduction (Rastegari et al., 2016). Such practical characteristics make BNN highly efficient for embedded devices (Ding et al., 2019) and friendly for memristor-based hardwares (Liu et al., 2020).

Despite the above-mentioned advantages of BNN, the severe performance drop hinders it from being widely deployed (Liu et al., 2020). Such a problem is particularly critical in binarized image SR, where dense pixel-wise predictions are required and the feature size is usually very large. The performance drop mainly comes from two parts: weights and activations binarization. **(1)** The weights are binarized from full-precision (*i.e.*, 32-bit) to 1-bit, being hard to extract high-frequency information. Even though the activations are full-precision, the SR output would still suffer from heavy degradation (Ma et al., 2019). **(2)** Binarizing activations (*i.e.*, features) would directly lose high-frequency information, which is the key component that SR networks try to recover. Moreover, after the computation operations between binarized weights and activations, the output would further lose pixel-wise detailed information with high uncertainty.

To address those issues, we propose a flexible residual binarization (FRB) technique for binarized image SR. **(1)** To tackle the first issue, we try to reduce the weight error with our second-order residual binarization (SRB). Specifically, we not only binarize the weights as a common practice, we further binarize weight residuals between 1-bit and full-precision weights. Such an SRB practice helps preserve network weight representation capability more effectively than direct binarization only. **(2)** Furthermore, to compensate the pixel-wise information loss, we propose Distillation-guided Binarization Training (DBT).

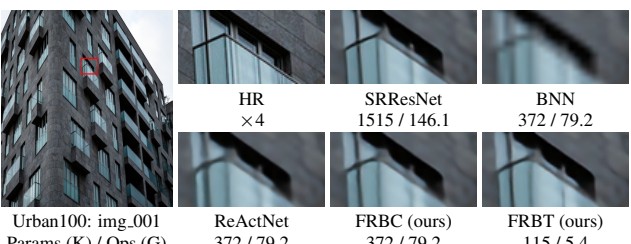

Figure 1: Visual samples of image SR ($\times 4$) by lightweight methods. SRResNet (Ledig et al., 2017) is a full-precision model and is used as a backbone for binarization by BNN (Courbariaux et al., 2016), ReActNet (Liu et al., 2020), and our FRBC. We also binarize SwinIR_S (Liang et al., 2021) and denote this version as FRBT. We provide the parameter (*i.e.*, Params (K)) and operation numbers (*i.e.*, Ops (G)). Input size is $3 \times 320 \times 180$ for Ops calculation.

Specifically, we try to transfer full-precision knowledge to narrow the representation content gap between the binarized and full-precision networks. A normalized attention form is built to uniformly align the contents of different bit-widths.

We further generalize our FRB to different types of networks and investigate its behaviors. Consequently, we apply our FRB to binarize CNN and Transformer based SR networks respectively, resulting in two binary baselines: FRBC and FRBT. Surprisingly, as shown in Fig. 1, our methods achieve promising results with comparable or much smaller computational resources.

Our main contributions are summarized as follows:

- We propose a simple yet effective method Flexible Residual Binarization (FRB) to accurately binarize full-precision image SR networks during the training.
- We propose an effective second-order residual binarization (SRB), which binarizes the image SR network with its weight residuals. SRB enhances the representation capacity of the binarized image SR network significantly for pixel-wise reconstruction.
- We propose Distillation-guided Binarization Training (DBT), which transfers full-precision knowledge to the binarized model. Specifically, we build a normalized attention form to uniformly align the contents of different bit-widths (*e.g.*, 32-bit and 1-bit).
- We employ our FRB to binarize CNN and Transformer based SR networks respectively, resulting in two binarized baselines: FRBC and FRBT. Our methods achieve superior performance over SOTA binarized SR methods quantitatively and visually.

## 2 RELATED WORK

### 2.1 LIGHTWEIGHT IMAGE SR

Lightweight image SR models have recently drawn more and more attention because of their resource-friendly properties. Usually, researchers pursue lightweight networks by architecture design, neural architecture search (NAS), knowledge distillation (KD), and network pruning. Ahn *et al.* constructed a cascading method upon a residual network (CARN) (Ahn et al., 2018). Hui *et al.* proposed an information multi-distillation network (IMDN) (Hui et al., 2019). Meantime, model compression methods have been introduced for lightweight SR, too. Chu *et al.* intorduced neural architecture search for image SR in FALSR (Chu et al., 2019). Knowledge distillation was employed to train lighter SR student networks (He et al., 2020; Lee et al., 2020). Such lightweight network designs and compression techniques have achieved promising performance. They either neglect the fine-grained parameter redundancy or consume a considerable number of additional computations.

### 2.2 MODEL QUANTIZATION

There are two main types of quantization methods: Post-Training Quantization (PTQ) and Quantization-Aware Training (QAT). PTQ has become increasingly popular due to its ability to quantize models without the need for retraining, resulting in numerous contributions in the field (Choukroun et al., 2019; Jhunjhunwala et al., 2021; Hubara et al., 2021; Li et al., 2021; Ding

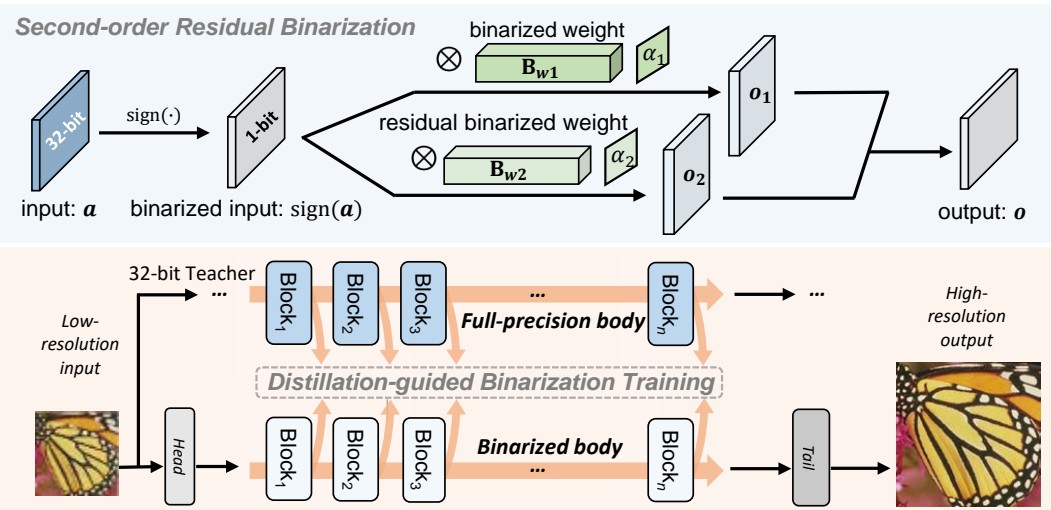

Figure 2: Overview of our Flexible Residual Binarization (FRB) for image SR networks. The upper (blue) is the *Second-order Residual Binarization*, where the SR network weights are binarized in a residual manner. The lower (orange) is the *Distillation-guided Binarization Training* that uniformly aligns the contents of different bit widths by constructing a normalized attention form.

et al., 2022). However, this approach only relies on limited expert knowledge and minimal GPU resources to calibrate the model, which significantly restricts its potential for achieving extreme low-bit quantization. Fortunately, QAT provides us with the opportunity to utilize the entire training pipeline to achieve aggressive low-bit quantization, including 1-bit binarization, and demonstrates promising performance (Martinez et al., 2020; Qin et al., 2020; Liu et al., 2020; 2018; Zhou et al., 2016; Courbariaux et al., 2016; Rastegari et al., 2016). This approach allows for more comprehensive model optimization, enabling the model to be trained to perform optimally in the quantized domain. QAT is usually seen as a powerful method for achieving extremely low-bit quantization.

Recent studies, including (Wang et al., 2020; Simons & Lee, 2019; Wang et al., 2022; Zhang et al., 2021; Qin et al., 2022), have demonstrated the effectiveness of 1-bit quantization, *i.e.*, binarization, as a highly efficient form of network quantization. This binarization technique compresses networks to achieve extreme computational and storage efficiency by using 1-bit binarized parameters. Compared to floating-point models, these quantized models significantly reduce computation resources and save time, and are hardware-friendly for edge devices.

### 2.3 BINARY NEURAL NETWORKS FOR IMAGE SR

Existing SR networks on resource-constrained devices are limited in usage by their high memory requirements and computational overhead. One major challenge is the heavy floating-point storage and operations involved in networks. Thus room for compression still exists from a bit-width perspective, which gives a strong motivation for the study of 1-bit binarized SR models (Xin et al., 2020; Jiang et al., 2021; Xia et al., 2022). Xin *et al.*designed a bit-accumulation mechanism to binarize full-precision SR networks (Xin et al., 2020). Xia *et al.*proposed a basic binary convolution unit for binarized image restoration (Xia et al., 2022). However, they mainly work on binarization for CNNs and lack the investigation about Transformer based binarized SR models.

### 3 FLEXIBLE RESIDUAL BINARIZATION FOR BINARIZED IMAGE SUPER-RESOLUTION

In this section, we first give an overview of binarization for single image super-resolution (SR) and raise the existing challenges of 1-bit SR networks. We then introduce our proposed flexible residual binarization (FRB) for image SR. Our FRB consists of two well-designed components: *Second-order Residual Binarization* (SRB) and *Distillation-guided Binarization Training* (DBT), which are designed for recovering the representation capacity and aligning the representation context, respectively. Afterward, we show how to utilize FRB for image SR and optimize the binary SR network (Fig. 2). We finally give more details about implementation.

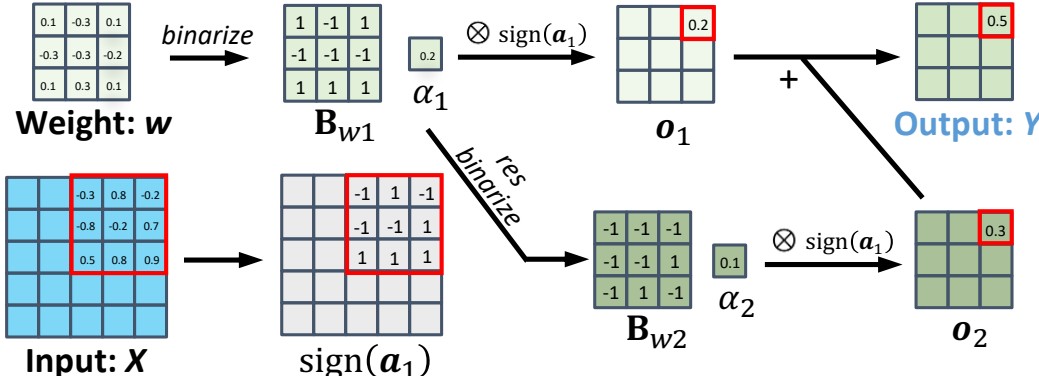

Figure 3: An computation example of our Second-order Residual Binarization (SRB). Our residual binarization allows the binarized weight representation to retain accurate information, further restoring the functionality of its full-precision counterpart compared to vanilla binarization. And the activation directly uses the sign function to binarize to avoid the extra burden during inference.

## 3.1 PRELIMINARIES: BINARIZATION IN SR

Here, we give a brief background to important key components in a general SR model binarization pipeline. Given a full-precision low-resolution (LR) image as input $I_{\text{LR}}$, the binary super-resolution network aims to obtain its full-precision high-resolution (HR) counterpart $I_{SR}$. We formulate such an image SR process with the neural network as follows

$$I_{SR} = \mathcal{F}_{\text{BSR}}(I_{\text{LR}}; \mathbf{\Theta}), \tag{1}$$

where $\mathcal{F}_{\text{BSR}}(\cdot)$ denotes the binary super-resolution (BSR) network with trainable parameters $\mathbf{\Theta}$. Specifically, we binarize the network $\mathcal{F}_{\text{BSR}}(\cdot)$ by the sign function, which is a standard choice for the task. The forward operation is the standard sign function,

$$\text{sign}(\boldsymbol{x}) = \begin{cases} 1 & \text{if } \boldsymbol{x} \geq 0 \\ -1 & \text{otherwise} \end{cases}. \tag{2}$$

Since this standard sign function is not continuous or differentiable, its backward operation can hardly achieved directly. Instead, the backward is replaced by the approximation,

$$\frac{\partial \, \text{sign}}{\partial \boldsymbol{x}} = \begin{cases} 1 & \text{if } |\boldsymbol{x}| \leq 1 \\ 0 & \text{otherwise} \end{cases}. \tag{3}$$

A floating-point precision weight matrix $\boldsymbol{w}$ can thus be binarized as,

$$\mathbf{B}_{\boldsymbol{w}} = \alpha \, \text{sign}(\boldsymbol{w}) \tag{4}$$

A scaling factor $\alpha$ is introduced to retain the magnitude of real-value weights. It is computed as

$$\alpha = \frac{1}{n} \boldsymbol{w}^{\top} \text{sign}(\boldsymbol{w}) = \frac{1}{n} \|\boldsymbol{w}\|_1. \tag{5}$$

After binarizing the SR networks, the storage size and computation can be significantly reduced due to the extremely reduced bit-width and highly efficient bitwise XNOR and bitcount operations (Rastegari et al., 2016). We then propose two techniques to improve binarized networks.

## 3.2 SECOND-ORDER RESIDUAL BINARIZATION FOR WEIGHT ERROR REDUCTION

While binarization promises reduced storage and faster inference, it substantially reduces the capacity of the original weights. It causes serious challenges for binarized image SR networks. This can be captured in the error caused by binarizing the continuous weights in Eq. 4 as,

$$\boldsymbol{\epsilon} = \boldsymbol{w} - \mathbf{B}_{\boldsymbol{w}}. \tag{6}$$

The error $\boldsymbol{\epsilon}$ represents the residual information that is lost in the binarization operation. Intuitively, we want to reduce this error. While this could be done by increasing the number of bits in the discrete representation, it does not allow for the use of efficient binary network operations.

In this work, we propose a different approach to reducing binarization errors. We perform a second-order binarization, in order to retrieve information lost in the error Eq. 6. This is performed by binarizing the error Eq. 6 and using it as a residual correction term to approximate continuous

weights. Our binarization strategy is thus expressed as,

$$\mathbf{B}_{\boldsymbol{w1}} = \alpha_1 \operatorname{sign}(\boldsymbol{w}), \qquad \alpha_1 = \frac{1}{n} \|\boldsymbol{w}\|_1, \qquad (7)$$

$$\mathbf{B}_{\boldsymbol{w2}} = \alpha_2 \operatorname{sign}(\boldsymbol{w} - \mathbf{B}_{\boldsymbol{w1}}), \ \alpha_2 = \frac{1}{n} \|\boldsymbol{w} - \mathbf{B}_{\boldsymbol{w1}}\|_1. \qquad (8)$$

We refer to $\mathbf{B}_{\boldsymbol{w1}}$ and $\mathbf{B}_{\boldsymbol{w2}}$ as the first and second order binarization, respectively. Note that the scaling factors are computed using the same formula Eq. 5.

In Eq. 7, the gradient estimation in the backward propagation for the sign function approximately follows Eq. 3. And for activation, the binarization operation follows the sign binarizer in Liu et al. (2020). Taking the binarized convolution unit as an example, the forward computation process of our second-order residual binarization (SRB) is expressed as,

$$o = \operatorname{sign}(\boldsymbol{a}) \otimes \mathbf{B}_{\boldsymbol{w1}} + \operatorname{sign}(\boldsymbol{a}) \otimes \mathbf{B}_{\boldsymbol{w2}}, \qquad (9)$$

where the $\otimes$ is the bitwise convolution consisting of XNOR and bitcount instructions (Arm, 2020; AMD, 2022). We also give an example of our technique in Fig. 3.

Second-order residual binarization (SRB) preserves the representation capability of weights better than direct binarization, while still being able to use bitwise instructions for efficient computation. Moreover, residuals enhance the representation capacity of binarized weights by making them closer to the original values and more diverse in the output space. Such a property can significantly boost the performance of binarized image SR networks.

## 3.3 DISTILLATION-GUIDED BINARIZATION TRAINING

In addition to the decrease in network representation capacity, the high discretization of binarization also leads to severe content distortion of representations. On the other hand, since most image SR models are composed block-by-block (Liang et al., 2021; Lim et al., 2017), for image SR networks, the $n$-block $\mathcal{F}_{\mathrm{BSR}}(\cdot)$ in Eq. 1 can be reformulated as,

$$I_{SR} = \mathcal{F}_{\mathrm{BSR}}(I_{\mathrm{LR}}; \boldsymbol{\Theta}) = \prod_{i=1}^{n} Blk_{\mathrm{BSR}_i}(I_{\mathrm{LR}}; \boldsymbol{\Theta}). \qquad (10)$$

Here, $Blk_{\mathrm{BSR}_i}$ denotes the $i$-th inner block of the SR network composed of several binarized computation units, including binarized convolution and linear units. Correspondingly, full-precision models and blocks is denoted as $\mathcal{F}_{\mathrm{SR}}(\cdot)$ and $Blk_{\mathrm{SR}_i}$. Lastly, $\prod$ denotes the composition of blocks.

Based on the above formulation and illustrations, the block-level ($k$-th block) representation distortion caused by binarization can be expressed as,

$$\mathcal{D}_k = \prod_{i=1}^{k} Blk_{\mathrm{SR}_i}(I_{\mathrm{LR}}; \boldsymbol{\Theta}) - \prod_{i=1}^{k} Blk_{\mathrm{BSR}_i}(I_{\mathrm{LR}}; \boldsymbol{\Theta}). \qquad (11)$$

To make the binarized SR model perform close to the full-precision level, intuitively, we should reduce the distortion $\mathcal{D}_i$ of each block in the model.

Therefore, we propose Distillation-guided Binarization Training (DBT) to align the representation content gap between binarized and full-precision SR networks (as Fig. 4). Inspired by (Martinez et al., 2020), we construct a normalized attention form for block-level representations to uniformly stabilize the contents in networks of different bit-widths. For example, the $i$-th block's formed representation in a binarized IR network can be formulated as

$$R_{\mathrm{BSR}_k} = \frac{\left( \prod_{i=1}^{k} Blk_{\mathrm{BSR}_i}(I_{\mathrm{LR}}; \boldsymbol{\Theta}) \right)^2}{\left\| \left( \prod_{i=1}^{k} Blk_{\mathrm{BSR}_i}(I_{\mathrm{LR}}; \boldsymbol{\Theta}) \right)^2 \right\|_{\ell 2}}, \qquad (12)$$

where $\| \cdot \|_{\ell 2}$ denotes the L2 normalization.

Then we distill full-precision representations to binarized ones. We target to consistently push binarized presentations to approach full-precision level representations:

$$\min \mathcal{L}_{\mathrm{DBT}} = \sum_{i=1}^{n} \hat{\mathcal{D}}_i = \sum_{i=1}^{n} \|R_{\mathrm{SR}_i} - R_{\mathrm{BSR}_i}\|_{\ell 2}. \qquad (13)$$

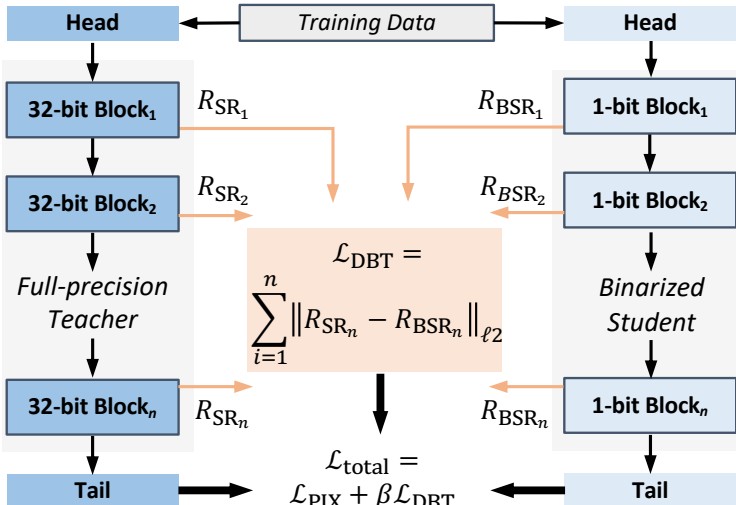

Figure 4: The computation flow of the loss function considering DBT. During training, the training data is simultaneously fed into the binarized SR network and its well-trained full-precision counterpart, and $\mathcal{L}_{\text{DBT}}$ is calculated according to the block-level intermediate representation (such as Eq. 13). In the end, $\mathcal{L}_{\text{DBT}}$ participates in the calculation of the total loss and jointly optimizes the binarized SR model with other loss items ($\mathcal{L}_{\text{PIX}}$ in Eq. 14).

Note that the binarized SR model and the full-precision replica are a pair of natural teachers and students because they have exactly the same architecture and significant differences in computation/storage. We highlight that this fact makes our DBT a flexible and architecture-generic technique, and the blockwise distillation implementation can even be fine-grained to a single computing layer level to suit various architectures. Such a property allows us to practice our compression techniques on various CNN- and Transformer-based image SR networks.

### 3.4 FRB FOR IMAGE SR

**Binarized Architectures.** For FRB, the SRB technique is allowed to be flexibly applied to various computational units in the architecture, such as convolutional and linear units. Therefore, for the image SR architecture using FRB, we apply SRB binarization to all computing units in the body part, which is the most computationally intensive, and maintain the full precision of the head and tail parts. In addition, the ReLU function is replaced by PReLU following Martinez et al. (2020).

**SR Model Training.** For the given training dataset $D = \left\{ I_{\text{LR}}^i, I_{\text{HR}}^i \right\}_{i=1}^K$ with $K$ low-resolution inputs and their corresponding HR counterparts, the image SR model with our proposed FRD is optimized by minimizing both the conventional pixel-wise $\mathcal{L}_{\text{PIX}}$ loss and $\mathcal{L}_{\text{DBT}}$ distillation loss:

$$\mathcal{L}_{\text{PIX}} = \frac{1}{n} \sum_{i=1}^k \left\| I_{\text{HR}}^i - I_{\text{SR}}^i \right\|_{\ell_1},$$

$$\mathcal{L}_{\text{total}} = \mathcal{L}_{\textbf{PIX}} + \beta \mathcal{L}_{\textbf{DBT}},$$

(14)

where the $\beta$ is a hyperparameter and is set as 1e-4 by default in our FRB, and $\mathcal{L}_{\textbf{DBT}}$ is in Eq. 13. Figure 4 also presents the computation flow of our training loss.

## 4 EXPERIMENTS

### 4.1 SETTINGS

**Data.** Following the common practice in image SR (Lim et al., 2017; Zhang et al., 2018a; Xin et al., 2020), we adopt DIV2K (Timofte et al., 2017) as the training data. Five benchmark datasets are used for testing: Set5 (Bevilacqua et al., 2012), Set14 (Zeyde et al., 2010), B100 (Martin et al., 2001), Urban100 (Huang et al., 2015), and Manga109 (Matsui et al., 2017).

**Evaluation.** To evaluate the reconstruction performance, we calculate PSNR and SSIM (Wang et al., 2004) values on the Y channel of the YCbCr space. For model complexity evaluation, we follow (Rastegari et al., 2016) and report the model size and operations of BNN. Specifically, we

| Method | Scale | Bits (W/A) | Set5 | | Set14 | | B100 | | Urban100 | | Manga109 | |
|---|---|---|---|---|---|---|---|---|---|---|---|---|
| | | | PSNR | SSIM | PSNR | SSIM | PSNR | SSIM | PSNR | SSIM | PSNR | SSIM |
| Bicubic | ×2 | -/- | 33.66 | 0.9299 | 30.24 | 0.8688 | 29.56 | 0.8431 | 26.88 | 0.8403 | 30.80 | 0.9339 |
| SRResNet | ×2 | 32/32 | 38.00 | 0.9605 | 33.59 | 0.9171 | 32.19 | 0.8997 | 32.11 | 0.9282 | 38.56 | 0.9770 |
| BNN | ×2 | 1/1 | 32.25 | 0.9118 | 29.25 | 0.8406 | 28.68 | 0.8104 | 25.96 | 0.8088 | 29.16 | 0.9127 |
| DoReFa | ×2 | 1/1 | 36.76 | 0.9550 | 32.44 | 0.9072 | 31.31 | 0.8883 | 29.26 | 0.8945 | 35.81 | 0.9682 |
| Bi-Real | ×2 | 1/1 | 32.32 | 0.9123 | 29.47 | 0.8424 | 28.74 | 0.8111 | 26.35 | 0.8161 | 29.64 | 0.9167 |
| IRNet | ×2 | 1/1 | 37.27 | 0.9579 | 32.92 | 0.9115 | 31.76 | 0.8941 | 30.63 | 0.9122 | 36.77 | 0.9724 |
| BAM | ×2 | 1/1 | 37.21 | 0.9560 | 32.74 | 0.9100 | 31.60 | 0.8910 | 30.20 | 0.9060 | N/A | N/A |
| BTM | ×2 | 1/1 | 37.22 | 0.9575 | 32.93 | 0.9118 | 31.77 | 0.8945 | 30.79 | 0.9146 | 36.76 | 0.9724 |
| ReActNet | ×2 | 1/1 | 37.26 | 0.9579 | 32.97 | 0.9124 | 31.81 | 0.8954 | 30.85 | 0.9156 | 36.92 | 0.9728 |
| BBCU-L | ×2 | 1/1 | 37.58 | 0.9590 | 33.18 | 0.9143 | 31.91 | 0.8962 | 31.12 | 0.9179 | 37.50 | 0.9746 |
| FRBC (ours) | ×2 | 1/1 | 37.71 | 0.9595 | 33.22 | 0.9141 | 31.95 | 0.8968 | 31.15 | 0.9184 | 37.90 | 0.9755 |
| FRBC+ (ours) | ×2 | 1/1 | 37.85 | 0.9600 | 33.32 | 0.9154 | 32.02 | 0.8977 | 31.29 | 0.9198 | 38.23 | 0.9762 |
| Bicubic | ×4 | -/- | 28.42 | 0.8104 | 26.00 | 0.7027 | 25.96 | 0.6675 | 23.14 | 0.6577 | 24.89 | 0.7866 |
| SRResNet | ×4 | 32/32 | 32.16 | 0.8951 | 28.60 | 0.7822 | 27.58 | 0.7364 | 26.11 | 0.7870 | 30.46 | 0.9089 |
| BNN | ×4 | 1/1 | 27.56 | 0.7896 | 25.51 | 0.6820 | 25.54 | 0.6466 | 22.68 | 0.6352 | 24.19 | 0.7670 |
| DoReFa | ×4 | 1/1 | 30.33 | 0.8601 | 27.40 | 0.7526 | 26.83 | 0.7104 | 24.29 | 0.7175 | 27.00 | 0.8470 |
| Bi-Real | ×4 | 1/1 | 27.75 | 0.7935 | 25.79 | 0.6879 | 25.59 | 0.6478 | 22.91 | 0.6450 | 24.57 | 0.7752 |
| IRNet | ×4 | 1/1 | 31.38 | 0.8835 | 28.08 | 0.7679 | 27.24 | 0.7227 | 25.21 | 0.7536 | 28.97 | 0.8863 |
| BAM | ×4 | 1/1 | 31.24 | 0.8780 | 27.97 | 0.7650 | 27.15 | 0.7190 | 24.95 | 0.7450 | N/A | N/A |
| BTM | ×4 | 1/1 | 31.43 | 0.8850 | 28.16 | 0.7706 | 27.29 | 0.7256 | 25.34 | 0.7605 | 29.19 | 0.8912 |
| ReActNet | ×4 | 1/1 | 31.54 | 0.8859 | 28.19 | 0.7705 | 27.31 | 0.7252 | 25.35 | 0.7603 | 29.25 | 0.8912 |
| BBCU-L | ×4 | 1/1 | 31.79 | 0.8905 | 28.38 | 0.7762 | 27.41 | 0.7303 | 25.62 | 0.7696 | 29.69 | 0.8992 |
| FRBC (ours) | ×4 | 1/1 | 31.83 | 0.8906 | 28.39 | 0.7763 | 27.41 | 0.7303 | 25.61 | 0.7693 | 29.71 | 0.8989 |
| FRBC+ (ours) | ×4 | 1/1 | 31.99 | 0.8927 | 28.48 | 0.7781 | 27.47 | 0.7319 | 25.73 | 0.7722 | 29.96 | 0.9018 |

Table 1: Quantitative results in CNN based binarized image SR methods. SRResNet is used as the full-precision backbone. Bits (W/A) denote the bits of weights and activations. The best and second best results are colored with red and cyan.

| Method | Scale | Bits (W/A) | Set5 | | Set14 | | B100 | | Urban100 | | Manga109 | |
|---|---|---|---|---|---|---|---|---|---|---|---|---|
| | | | PSNR | SSIM | PSNR | SSIM | PSNR | SSIM | PSNR | SSIM | PSNR | SSIM |
| SwinIR_S | ×2 | 32/32 | 38.14 | 0.9611 | 33.86 | 0.9206 | 32.31 | 0.9012 | 32.76 | 0.9340 | 39.12 | 0.9783 |
| FRBT (ours) | ×2 | 1/1 | 37.69 | 0.9594 | 33.24 | 0.9148 | 31.96 | 0.8968 | 31.13 | 0.9184 | 37.90 | 0.9753 |
| FRBT+ (ours) | ×2 | 1/1 | 37.82 | 0.9598 | 33.32 | 0.9156 | 32.02 | 0.8976 | 31.26 | 0.9197 | 38.23 | 0.9762 |
| SwinIR_S | ×4 | 32/32 | 32.44 | 0.8976 | 28.77 | 0.7858 | 27.69 | 0.7406 | 26.47 | 0.7980 | 30.92 | 0.9151 |
| FRBT (ours) | ×4 | 1/1 | 31.79 | 0.8896 | 28.35 | 0.7757 | 27.41 | 0.7306 | 25.55 | 0.7681 | 29.68 | 0.8988 |
| FRBT+ (ours) | ×4 | 1/1 | 31.92 | 0.8913 | 28.43 | 0.7774 | 27.47 | 0.7320 | 25.65 | 0.7704 | 29.92 | 0.9016 |

Table 2: Quantitative results in Transformer based binarized image SR methods. We use SwinIR_S as the backbone. We find quantization of Transformer models causes a significant quality loss. This is an interesting problem for future work.

calculate the BNN parameters via $Params\_1 = Params\_f/32$, where $Params\_f$ is the full-precision counterpart parameters. We calculate BNN operations via $Ops\_1 = Ops\_f/64$, where $Ops\_f$ denotes operations of the full-precision counterpart. Based on $Params\_1$ and $Ops\_1$, we further provide theoretical compression ratios for parameters and operations.

**Proposed Binary Baselines.** We apply our FRB to binarize CNN and Transformer based image SR baselines. Specifically, following BAM (Xin et al., 2020) and BTM (Jiang et al., 2021), we use SRResNet (Ledig et al., 2017) as CNN SR backbone, binarize its body part, and name this version as FRBC. We further generalize our FRB to a lightweight Transformer SR backbone, SwinIR_S (Liang et al., 2021). We binarize SwinIR_S and name this version as FRBT. In addition, we use self-ensemble (Lim et al., 2017) to further enhance them and denote as FRBC+ and FRBT+.

**Training Strategy.** In the training phase, same as previous work (Lim et al., 2017; Zhang et al., 2018a; Xin et al., 2020; Liang et al., 2021), we conduct data augmentation (random rotation by 90°, 180°, 270° and horizontal flip). We train the model for 300K iterations. Each training batch extracts 32 image patches, whose size is 64×64. We utilize Adam optimizer (Kingma & Ba, 2015) ($\beta_1$=0.9, $\beta_2$=0.999, and $\epsilon$=$10^{-8}$) during training. The initial learning rate $2\times10^{-4}$, which is reduced by half at the $250K$-th iteration. PyTorch (Paszke et al., 2017) is employed to conduct all experiments.

## 4.2 MAIN COMPARISONS

For CNN-based image SR networks, we choose SRResNet (Ledig et al., 2017) as the backbone. We then adopt different binary methods: BNN (Courbariaux et al., 2016), DoReFa (Zhou et al., 2016), Bi-Real (Liu et al., 2018), IRNet (Qin et al., 2020), BAM (Xin et al., 2020), BTM (Jiang et al., 2021), ReActNet (Liu et al., 2020), BBCU-L (Xia et al., 2022), and our FRBC.

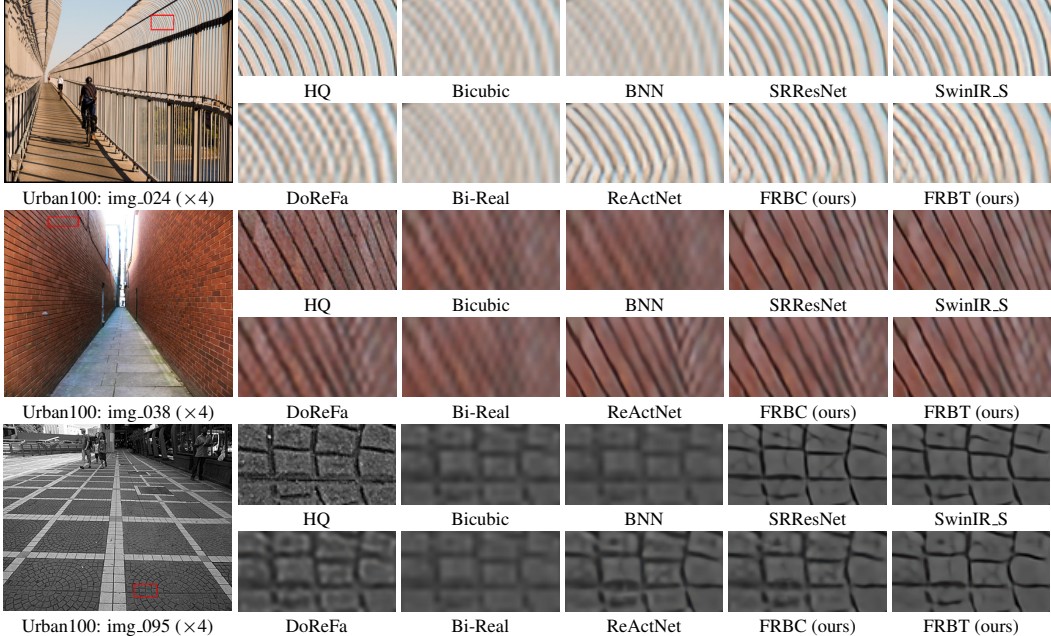

Figure 5: Visual comparison (×4) with lightweight and binarized image SR networks on Urban100 dataset. SRResNet and SwinIR_S are full-precision and used as references. Our FRBC performs better than other binarized methods with the same backbone SRResNet.

**Quantitative Results.** In Tab. 1, we provide Params, Ops, PSNR, and SSIM values. When using the same CNN-based backbone SRResNet, our FRBC achieves comparable or better PSNR/SSIM scores with similar number of Params and Ops as others.

**Generalize to Transformer.** For Transformer-based image SR networks, we choose the lightweight SwinIR_S (Liang et al., 2021) as the backbone. Due to the more challenging case in Transformer binarization and the performance observation in CNN-based methods, we only apply our FRB to binarize SwinIR_S as FRBT. We further provide results of our binarized Transformer baseline, FRBT. In Tab. 2, we can see FRBT reduces the Params and Ops obviously. But the performance gap between FRBT and SwinIR_S is larger than that between FRBC and SRResNet. It means that it is more challenging to binarize Transformer-based image SR networks. However, we investigate firstly the binary behavior in the image SR Transformer. We open the way to further improve binarization performance and narrow the performance gap between the binary and full-precision models.

**Compression Ratio.** In Tab. 3, we provide the compression ratio and speedup in terms of Params and Ops respectively. We quantize full-precision networks, SRResNet and SwinIR_S, which are stored with data type single precision floating point. Their model size

| Method | Bits (W/A) | Params (K) (↓ Compr. Ratio) | Ops (G) (↓ Compr. Ratio) | Urban100 PSNR | Urban100 SSIM |
|---|---|---|---|---|---|
| SRResNet | 32 / 32 | 1367 (0%) | 85.4 (0%) | 32.11 | 0.9282 |
| FRBC (ours) | 1 / 1 | 225 (↓ 83.5%) | 18.6 (↓ 78.2%) | 31.15 | 0.9184 |
| SwinIR_S | 62 / 32 | 910 (0%) | 62.4 (0%) | 32.76 | 0.9340 |
| FRBT (ours) | 1 / 1 | 95 (↓ 89.6%) | 4.3 (↓ 93.1%) | 31.13 | 0.9184 |

Table 3: Compression ratio of SRResNet and SwinIR_S (×2). Bits (W/A) denote the weights and activations bit number. We set the input size as 3×320×180 for Ops calculation.

(*i.e.*, Params) and operations (*i.e.*, Ops) can be reduced considerably. Following BBCU-L (Xin et al., 2020), we only binarize the weights and activations in the body part module. But, we calculate the compression ratio and speedup over the whole model. Our FRBC and FRBT still achieve around 80% compression ratio. The reconstruction performance could drop, but binary quantization can significantly save the model size and operations.

**Visual Results.** In Fig. 5, we provide visual results of representative and recently leading methods with scale ×4 in terms of some challenging cases. For each case, we compare with several BNN methods, like BNN, DoReFa, Bi-Real, and ReActNet. Our FRBC obtains obviously better results than theirs on the same CNN-based SR backbone. We further consider full-precision models (*i.e.*, SRResNet and SwinIR_S) and their corresponding binary counterparts (*i.e.*, FRBC and FRBT). Their

visual difference is small. These visual comparisons further demonstrate the effectiveness of our FRBC and FRBT, which is consistent with the observations in Tabs. 1 and 2.

### 4.3 ABLATION STUDY

To demonstrate the effectiveness of our contributions, we conduct ablation studies about second-order residual binarization (SRB) and Distillation-guided Binarization Training (DBT). To save training time and resources, we reduce the input size to 48×48 and train 200K iterations. We use SRResNet (Ledig et al., 2017) as the image SR backbone. We use the well-known and basic binary method DoReFa (Zhou et al., 2016) as a baseline. We then equip SRB or/and DBT to SRResNet and binarize it. We report PSNR/SSIM values on B100, Urban100, and Manga109 in Tab. 4.

**Second-order Residual Binarization (SRB).** As a vanilla version of binary method, DoReFa (Zhou et al., 2016) has shown the basic SR performance. We conduct second-order residual binarization (SRB) for the weights in the computation unit. In Tab. 4, we can see our proposed SRB significantly boosts the performance

| Method | B100 | | Urban100 | | Manga109 | |
|---|---|---|---|---|---|---|
| | PSNR | SSIM | PSNR | SSIM | PSNR | SSIM |
| DoReFa | 31.25 | 0.8873 | 29.15 | 0.8929 | 35.66 | 0.9676 |
| SRB | 31.77 | 0.8939 | 30.56 | 0.9113 | 37.51 | 0.9739 |
| DBT | 31.26 | 0.8873 | 29.18 | 0.8929 | 35.77 | 0.9678 |
| FRB (*i.e.*, URB+DBT) | 31.83 | 0.8948 | 30.74 | 0.9138 | 37.64 | 0.9744 |

Table 4: Ablation study (×2) about our proposed second-order residual binarization (SRB), Distillation-guided Binarization Training (DBT), and flexible residual binarization (FRB). The SR backbone is SRResNet (Ledig et al., 2017).

of the binary network and reduces the performance drop. Our SRB achieves around 0.4∼1.8 dB and 0.0066∼0.0184 in terms of PSNR and SSIM. In image SR, residual learning or residual feature usually extracts high-frequency information, which contributes much to high-quality reconstruction. On the other hand, feature size usually is very large or has an arbitrary size, which consumes lots of computational resources. Instead, we turn to enhancing the representation capacity with SRB. This performance gain from SRB over DoReFa indicates that binarizing weights residually is an efficient way to reduce the performance gap in binary SR models.

**Distillation-guided Binarization Training (DBT).** During the network training, there are still full-precision weights for binarization. It is straightforward to utilize full-precision information as guidance. As shown in Tab. 4, using DBT would only increase the performance by marginal gains, except for Manga109 (*i.e.*, 0.11 dB PSNR gain). Such an observation gives us two thoughts. **(1)**. Our proposed DBT is effective to boost the binary SR performance independently. This is mainly because DBT leads to better representation content alignment in the image SR process. **(2)**. Knowledge distillation can hardly achieve notable gains without considering the specific property of image super-resolution (SR). Then, we are inspired to jointly integrate SRB and DBT together by aiming to reconstruct more high-frequency information effectively.

**Flexible Residual Binarization (FRB).** When we jointly train the SR network with reconstruction and distillation losses, we reach flexible residual binarization (FRB). Considering the whole data lines in Tab. 4, we find that FRB achieves even higher performance over the vanilla binary baseline DoReFa (Zhou et al., 2016), resulting in larger gains than those obtained by using SRB and DBT independently. These observations demonstrate that our FRB can well extract more valuable information (*i.e.*, high-frequency information) with residual binarized weights and also transfer full-precision knowledge to the binary image SR network.

## 5 CONCLUSION

In this work, we propose a flexible residual binarization (FRB) technique to dramatically reduce the parameters and operations of full-precision image super-resolution (SR) networks. To extract more high-frequency information for better image reconstruction, we propose a second-order residual binarization (SRB). Our proposed SRB binarizes the residual weights, which has been demonstrated to be pretty effective over binarizing weights directly. At the same time, to make the binarized SR model perform closer to its full-precision counterpart, we transfer full-precision knowledge to guide the training of binary SR networks. Specifically, we propose Distillation-guided Binarization Training (DBT), which uniformly aligns the contents of different bit-widths. We finally apply our FRB to binarize both CNN and Transformer based SR methods, resulting in two baselines: FRBC and FRBT. We conduct extensive ablation studies and main experiments to show the effectiveness of our proposed components. Surprisingly, we find that FRBT obtains comparable or even better performance than FRBC with much fewer Params and Ops. To this end, our FRB opens the way to compress BNNs with efficient hardware, like FPGA, CPU, and GPU.

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
