# SUPPLEMENTARY MATERIALS: FLEXIBLE RESIDUAL BINARIZATION FOR IMAGE SUPER-RESOLUTION

## 1  MORE ANALYSES ABOUT MOTIVATION

We give more detailed analyses of our motivation for why we propose flexible residual binarization (FRB) in terms of weights instead of activations.

**Magnitude**. In the binarization, we also have to record the magnitude $\alpha$. If we want to binarize activations (*i.e.*, features), we have to calculate a $\alpha$ for each activation. Such a dynamic magnitude calculation causes lots of additional operations. Instead, to binarize each weight, we only have to compute the magnitude once, which is shared by different activations and saves computations dramatically.

**Tensor Size**. In image super-resolution (SR) or even other image restoration tasks, the input would have an arbitrary size. In some cases, the input size would be pretty large and lead to huge computational costs. However, the weights have fixed sizes, which need fixed operations during residual computations.

## 2  COMPETITIVE SOLUTION COMPARISON

As stated in the method part of the main paper, we not only binarize weights first but also binarize the residual weights. One might raise a question about the difference between our residual binarization and 2-bit quantization. We clarify it by their different implementations in practice.

Binarization (*i.e.*, 1-bit quantization) can be computed with XNOR and bitcount efficiently. While 2-bit quantization cannot be computed in this way. Instead, it is computed with integer operators, which is far less efficient than the 1-bit implementation.

## 3  NETWORK WEIGHT VISUALIZATION

In Fig. 1, we visualize weights from full-precision, vanilla binarized, and our FRB. Weight visualization with lightweight full-precision and binarized image SR networks. SRResNet is used as the full-precision backbone. Specifically, we extract the weights from the first convolutional (Conv) layer of the first residual block. The beginning 12 channels are used for evaluation due to the limited space. For each output channel, we arrange kernels (size: $3\times3\times64$) into 2-D ones with the size of $24\times24$.

We first provide the weight differences in Tab. 1. Specifically, for each channel weight, we sum the absolute difference between full-precision and binarized weight, and then divide the sum of L1 norm of full-precision weight. We can see our FRB obtains smaller differences to the full-precision than vanilla binarization. We also calculate the our improvement over vanilla binarization. As shown in Tab. 1, our FRB achieves **22.32%** ∼ **44.10%** improvements and further narrows the weight degradation.

We further provide two groups of visualization. In each group, we show the full-precision weights from different channels in the first row. Then, we provide the vanilla binarized weights (*i.e.*, DoReFa) in the second row. We finally show the binarized weights with our proposed flexible residual binarization (FRB) method. We can see the weights produced by our FRB look more similar to full-precision ones than those with vanilla binarization.

| Channel Index | 1 | 2 | 3 | 4 | 5 | 6 | 7 | 8 | 9 | 10 | 11 | 12 |
|---|---|---|---|---|---|---|---|---|---|---|---|---|
| Vanilla Binarized | 0.6510 | 0.6679 | 0.5942 | 0.6362 | 0.6634 | 0.6199 | 0.6351 | 0.6098 | 0.6304 | 0.6221 | 0.6096 | 0.6168 |
| FRB (ours) | 0.4121 | 0.4536 | 0.3321 | 0.4661 | 0.5153 | 0.4475 | 0.3644 | 0.3637 | 0.3752 | 0.4291 | 0.4040 | 0.4195 |
| Improvement (%) | 36.69 | 32.09 | 44.10 | 26.73 | 22.32 | 27.82 | 42.62 | 40.35 | 40.47 | 31.02 | 33.73 | 31.99 |

Table 1: The weight differences between full-precision model and binarized ones (*i.e.*, Vanilla Binarized and our FRB). Here, we only provide the differences for the beginning 12 feature channels from the first Conv in the first residual block.

## 4 MORE VISUAL RESULTS

For CNN-based image SR networks, we choose SRResNet (Ledig et al., 2017) as the backbone. We then adopt different binary methods: BNN (Courbariaux et al., 2016), DoReFa (Zhou et al., 2016), Bi-Real (Liu et al., 2018), IRNet (Qin et al., 2020), BAM (Xin et al., 2020), BTM (Jiang et al., 2021), ReActNet (Liu et al., 2020), and our FRBC. For Transformer-based image SR networks, we choose the lightweight SwinIR_S (Liang et al., 2021) as the backbone. Due to the more challenging case in Transformer binarization and the performance observation in CNN-based methods, we only apply our FRB to binarize SwinIR_S as FRBT. We further provide results of our binarized Transformer baseline, FRBT.

We provide more visual results in Figs. 2, 3, and 4. We can see that our proposed FRBC performs comparable or better than other compared methods in most cases. When it comes to more challenging cases, our proposed FRBT obtains the best reconstruction quality. When comparing FRBT and its corresponding full-precision model SwinIR_S, we find that the differences between them are limited. Those observations further demonstrate the effectiveness of the proposed method.

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

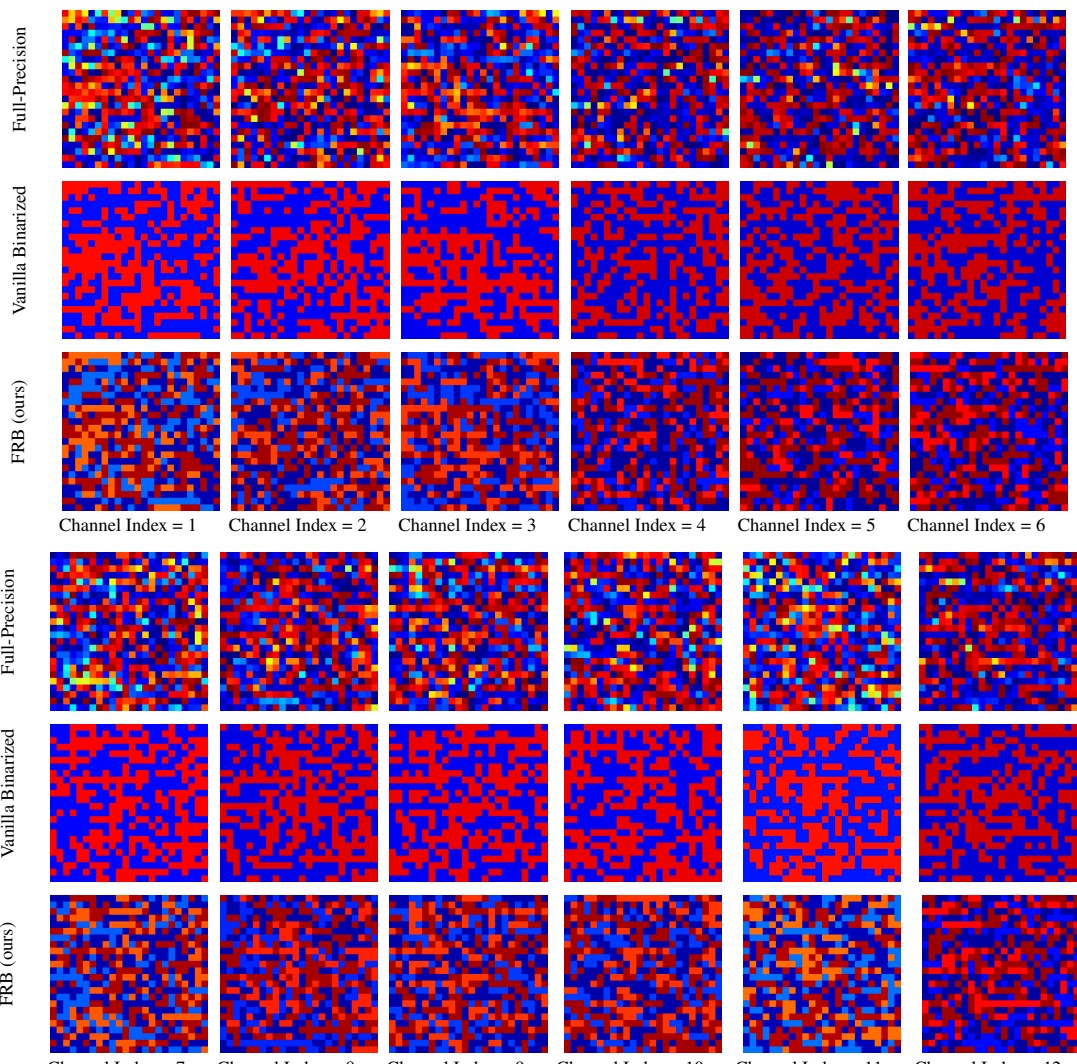

Figure 1: Weight visualization with lightweight full-precision and binarized image SR networks. SRResNet is used as the full-precision backbone. We provide two groups of visualization. In each group, we show the full-precision weights from different channels in the first row. Then, we provide the vanilla binarized weights (*i.e.*, DoReFa) in the second row. We finally show the binarized weights with our proposed flexible residual binarization (FRB) method.

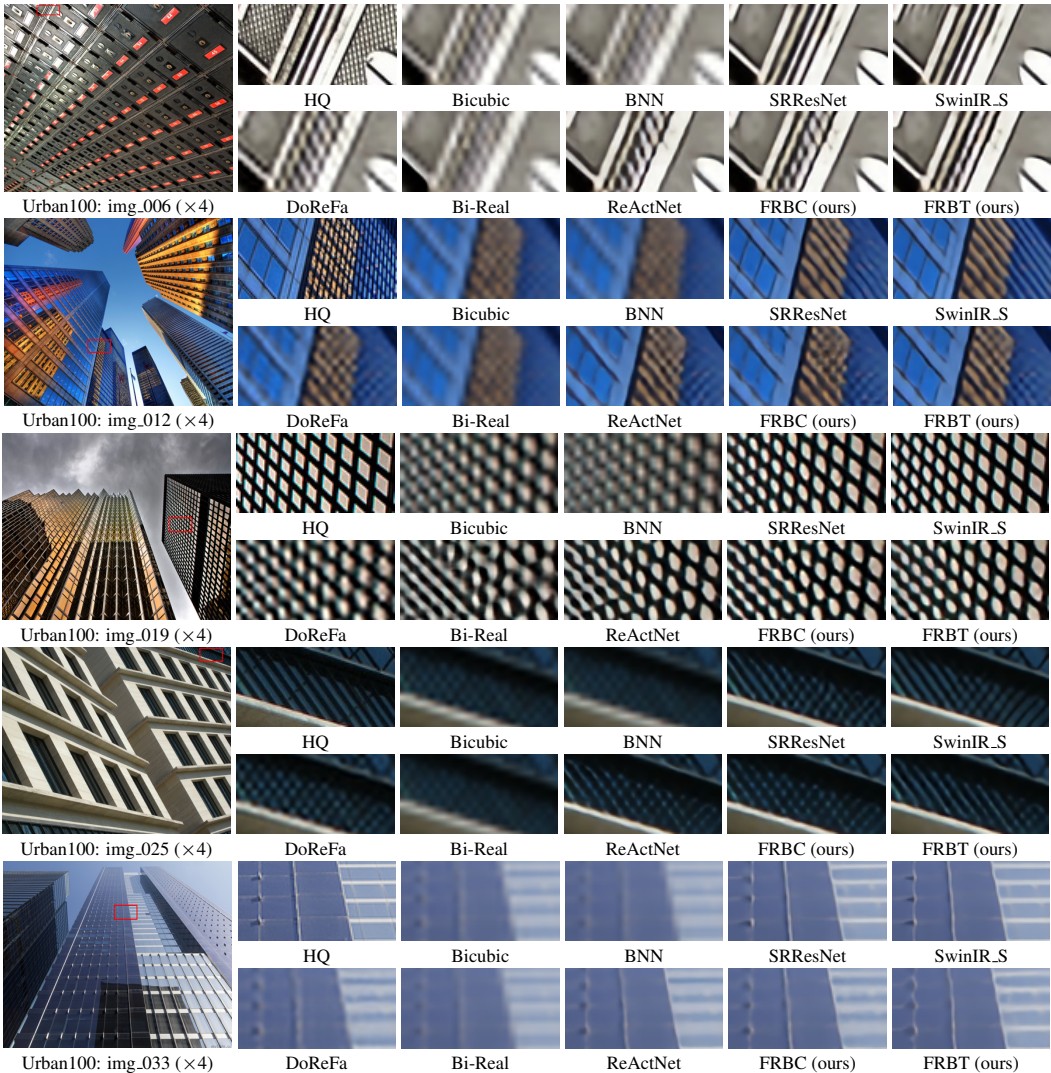

Figure 2: Visual comparison (×4) with lightweight and binarized image SR networks on Urban100 dataset. SRResNet and SwinIR_S are full-precision and used as references. Our FRBC performs better than other binarized methods with the same backbone SRResNet. Our Transformer baseline FRBT obtains results close to SwinIR_S.

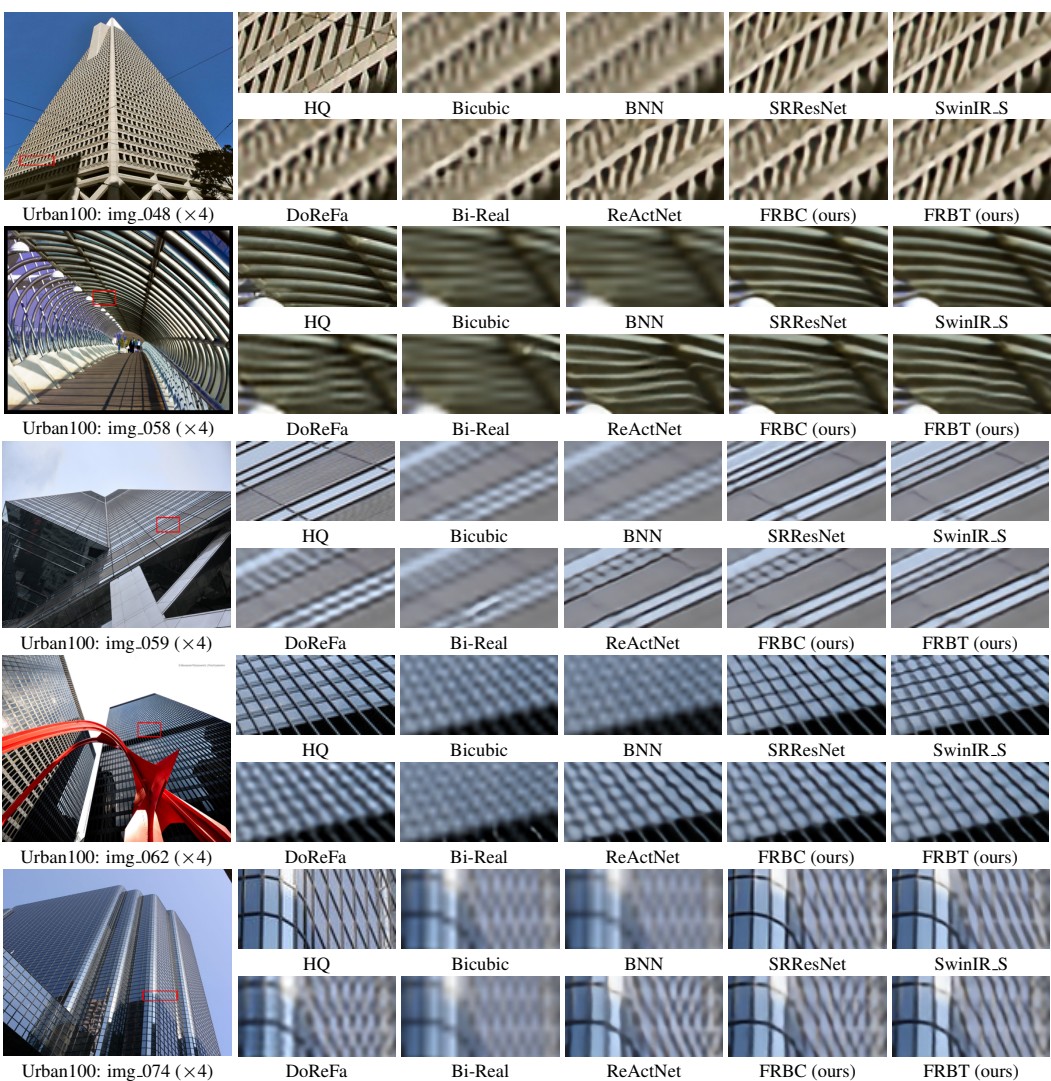

Figure 3: Visual comparison (×4) with lightweight and binarized image SR networks on Urban100 dataset. SRResNet and SwinIR_S are full-precision and used as references. Our FRBC performs better than other binarized methods with the same backbone SRResNet. Our Transformer baseline FRBT obtains results close to SwinIR_S.

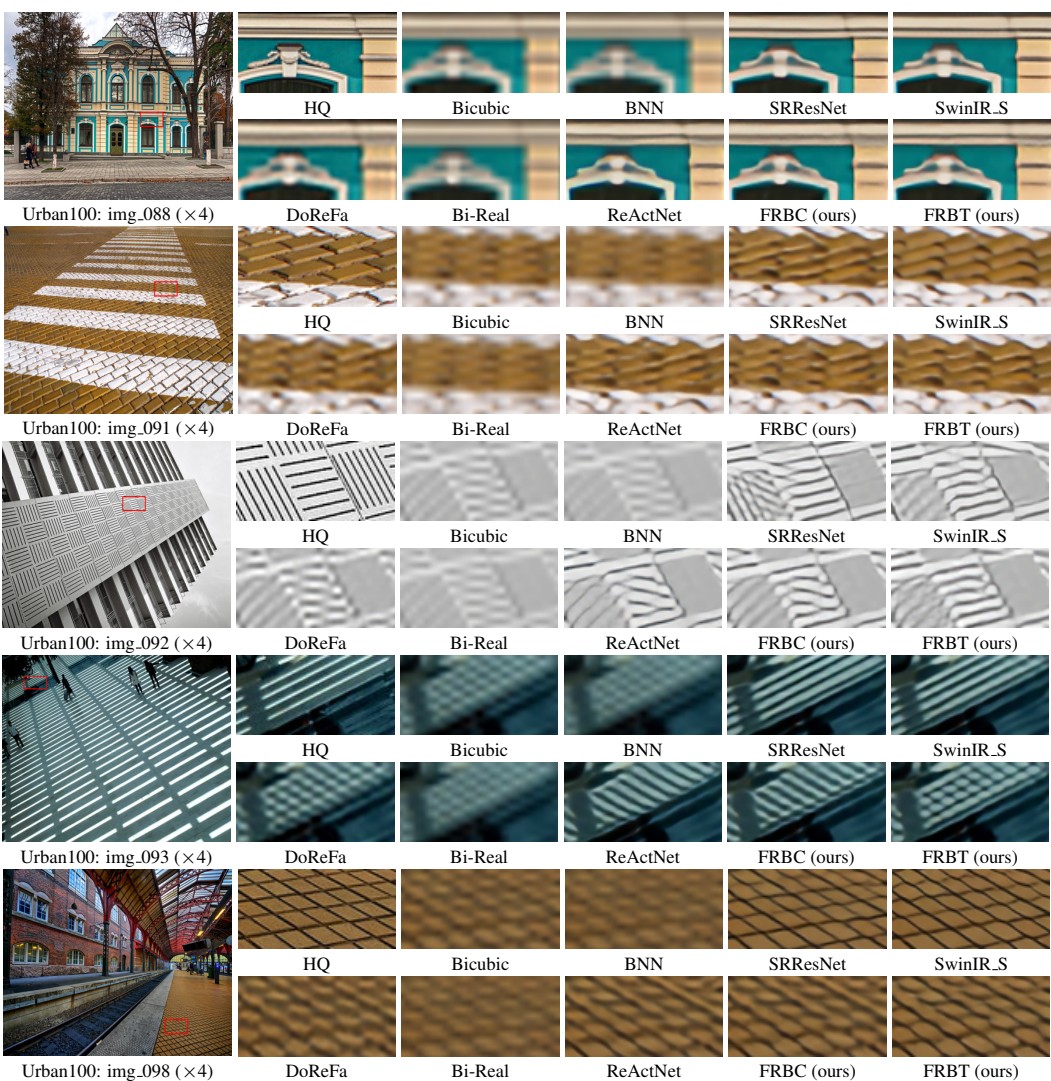

Figure 4: Visual comparison (×4) with lightweight and binarized image SR networks on Urban100 dataset. SRResNet and SwinIR_S are full-precision and used as references. Our FRBC performs better than other binarized methods with the same backbone SRResNet. Our Transformer baseline FRBT obtains results close to SwinIR_S.