# OpenReview forum: "Flexible Residual Binarization for Image Super-Resolution"
_ICLR.cc/2024/Conference — Submitted to ICLR 2024_

### Official Review · Reviewer_jLFb · 2023-10-24

**Soundness:** 3 good
**Presentation:** 3 good
**Contribution:** 3 good
**Rating:** 8
**Confidence:** 4

**Summary:**

In this paper, the authors propose a network binarization method for image SR networks. Specifically, the authors develop a second-order residal binarization method to reduce the binarization error. In addition, a distillation-guided binarization training strategy is introduced to transfer knowledge from the full-precision network to the binarized one. Experiments show that the proposed method outperforms previous network binarization methods on benchmark datasets.

**Strengths:**

- The proposed method technically sounds.
- This paper is well written and easy to follow.

**Weaknesses:**

- The idea of residual binarization and knowledge from full-precision models to binarized ones has been studied in other tasks. From this point of view, the technical contribution of this paper is limited.
- The residual binarization inevitably introduces additional parameters and such overhead should be discussed.
- Following the above comment, I wonder whether higher-order binarization (larger than 2) is able to introduce further performance gains and the optimal order to achieve a balance between accuracy and efficiency.
- It is recommended to compare the proposed knowledge distillation method with previous ones to demonstrate its superiority in transferring knowledge from full-precision models to binarized ones.

**Questions:**

please see weaknesses

---

> ### Author Response · Authors · 2023-11-16
> **Response to Reviewer jLFb (1/2)**
>
> We thank the reviewer for the constructive feedback and comments. We respond to the concerns below:
>
> > **Q1:** The idea of residual binarization and knowledge from full-precision models to binarized ones has been studied in other tasks. From this point of view, the technical contribution of this paper is limited.
>
> **A1:** We clarify that our proposed SRB in FRB differs from existing residual quantization techniques in significantly improving accuracy while reducing additional inference burden.
>
> Some existing works, such as HORQ [ref1], employ multi-group binarization of base linear approximations of original activations to recover information from activations directly. However, as activations are dynamic in each inference, high-order residual operations on activations in SR networks introduce significant floating-point computational burdens during the inference process. Therefore, our approach involves residual binarization of fixed intermediate activations during inference, avoiding the floating-point multiplications caused by residual operations (as shown in Eq. 9). Additionally, we constrain the order of residuals on weights in SRB to the second order, preventing marginal diminishing returns on the performance enhancement of binary SR by higher-order residuals and unnecessary efficiency decline.
>
> [ref1] Performance Guaranteed Network Acceleration via High-Order Residual Quantization. ICCV 2017.
>
> > **Q2:** The residual binarization inevitably introduces additional parameters and such overhead should be discussed.
>
> **A2:** We clarify that FRB brings little overhead in exchange for significant accuracy improvements and experimentally show that FRB can bring improved performance with fewer parameters.
>
> In Tab. A2, we compare the Ops and Params of our FRBC with the other methods. Our FRB achieves significant improvements with little overhead increases. We retrain our FRBC with scale x2 to keep much fewer parameters than other methods by removing one residual block (RB) in SRResNet, namely 15 RBs. We denote this version as FRBC-S. The results of FRBC-S still significantly outperform other methods.
>
> Moreover, we also provide the inference time on ARM64 CPU Raspberry Pi 4 Model B of different binarization methods with the same backbone SRResNet. FRBC achieves significant PSNR gains on Set5 (×2) with less than 0.02% increased inference time. The time of 15-RB FRBC-S is less than the existing methods while achieving higher accuracy.
>
> | Methods | FP32 | BNN | DoReFa | Bi-Real | IR-Net | BAM | BTM | ReActNet | FRBC | FRBC-S |
> | - | - | - | - | - | - | - | - | - | - | - |
> | Time (s) | 2.91 | 0.62 | 0.63 | 0.63 | 0.63 | 0.66 | 0.64 | 0.63 | 0.66 | 0.59 |
> | Ops (G)     | 85.4 | 18.6 | 18.6 | 18.6 | 18.6 | 18.6 | 18.6 | 18.6 | 19.7 | 17.5 |
> | Params (K) | 1367 | 225 | 225 | 225 | 225 | 225 | 225 | 225 | 238 | 223 |
> | PSNR | 38.00 | 32.25 | 36.76 | 37.27 | 32.32 | 37.21 | 37.22 | 37.26 | 37.71 | 37.68 |
>
> Table A2: Quantitative comparison (x2). We use 15 RBs in our FRBC-S. Input size is 3x320x180 for Ops calculation.
>
> > **Q3:** Following the above comment, I wonder whether higher-order binarization (larger than 2) is able to introduce further performance gains and the optimal order to achieve a balance between accuracy and efficiency.
>
> **A3:** We clarify that we constrain the order of residuals on weights in SRB to the second order to prevent marginal diminishing returns on the performance enhancement of binary SR by higher-order residuals and unnecessary efficiency decline. We also follow the reviewer's suggestions to evaluate the FRB with higher-order residual binarization in Table A3. When we quantize the SR model to third order, we find that the accuracy gain is tiny while the computation burden increases.
>
> | Methods | Params | Ops | Set5 | Set14 | B100 | Urban100 | Manga109 |
> | - | - | - | - | - | - | - | - |
> | ReActNet             | 225 | 18.6 | 37.26 | 32.97 | 31.81 | 30.85 | 36.92 |
> | FRB (ours)           | 238 | 19.7 | 37.71 | 33.22 | 31.95 | 31.15 | 37.90 |
> | FRB (third-order) | 255 | 23.0 | 37.72 | 33.24 | 31.95 | 31.45 | 37.91 |
>
> Table A3. Compared with FRB, the accuracy improvement of binarization extended to the third-order residual is limited, while the burden is more significant.

---

> > ### Author Response · Authors · 2023-11-16
> > **Response to Reviewer jLFb (2/2)**
> >
> > > **Q4:** It is recommended to compare the proposed knowledge distillation method with previous ones to demonstrate its superiority in transferring knowledge from full-precision models to binarized ones.
> >
> > **A4:** We follow the reviewer’s suggestion to evaluate the previous (vanilla) distillation further and compare it to our DBT. Table A4 shows vanilla distillation, taking l2-norm to the representation in Eq. (11) without the attention form. The results show that the proposed DBT outperforms vanilla distillation significantly, and the latter even causes crashes. Thus, the results indicate the necessity of the DBT form in our FRB.
> >
> > | Methods | Set5 | Set14 | B100 | Urban100 | Manga109 |
> > | - | - | - | - | - | - |
> > | Vanilla KD | 36.04 | 32.08 | 31.04 | 28.81 | 34.71 |
> > | DBT (ours) | 37.71	| 33.22 | 31.95 | 31.15 | 37.90 |
> >
> > Table A4: Quantitative comparison of distillation (scale ×2).

---

> > > ### Comment · Reviewer_jLFb · 2023-11-22
> > >
> > > Thank the authors for the response. Currently, I lean to accept this paper as the rebuttal has addressed most of my concerns. It is recommended to include these additional discussion and analyses in the main paper or the supplemental material.

---

### Official Review · Reviewer_iE9s · 2023-10-29

**Soundness:** 3 good
**Presentation:** 4 excellent
**Contribution:** 3 good
**Rating:** 8
**Confidence:** 5

**Summary:**

This paper proposes a binary neural network for super-resolution tasks. The main motivation is to improve the residual binarization of high-frequency information through the residual binarization method and further improve the performance with distillation technology. This paper is well-motivated, and the results show that the proposed method performs better than existing binary SR models.

**Strengths:**

(1) The paper is well-motivated. Binarization can help compress and speed up SR models. The comparison results given in the paper show that compared with the original SR model, the acceleration and compression of the binary SR model are significant. The binary SR model can also be closer to practicality by improving the accuracy after binarization.

(2) Reveals the nature of the performance degradation of the binary SR model. The author realizes that binary compression causes the loss of high-frequency features in the SR model, which is important for SR tasks. Therefore, the author improves the representation capability of features through residual binarization, and the accuracy rate is significantly improved.

(3) The proposed residual binarization technology is efficient. I think this technology attempts to use a balanced trade-off to solve the performance bottleneck of the binary SR model and maintain the acceleration properties of the bitwise operations of the computing unit (as described in Eq. 9). Since heads and tails are generally not quantified, the binary SR model using this technique adds negligible computation yet significantly improves accuracy.

(4) The binary distillation method is effective and versatile. The distillation between binarized and full-precision networks is natural since they have almost the same architecture. This allows distillation methods to run well on CNN and transformer-based models.

**Weaknesses:**

(1) What are the advantages of binarization compared to other compression methods, such as quantization (multi-bit) and distillation (without binarization)? Binarization seems to cause a more significant performance degradation. I suggest the author compare the proposed method with existing multi-bit quantization methods and discuss the advantages and disadvantages compared with other compression methods.

(2) The authors claim that residual binarization reduces the binarization error, which is intuitive. However, are there any quantitative results that show this?

(3) What is the acceleration effect on actual hardware? The author only discusses the reduction in FLOPs of binary SR networks. I am curious about the acceleration on real hardware.

**Questions:**

See weakness.

---

> ### Author Response · Authors · 2023-11-16
> **Response to Reviewer iE9s**
>
> We thank the reviewer for the constructive and helpful suggestions. We provide additional discussions below:
>
> > **Q1:** What are the advantages of binarization compared to other compression methods, such as quantization (multi-bit) and distillation (without binarization)? Binarization seems to cause a more significant performance degradation. I suggest the author compare the proposed method with existing multi-bit quantization methods and discuss the advantages and disadvantages compared with other compression methods.
>
> **A1:** We follow the reviewer's suggestions to compare binarization for SR tasks with other compression methods:
>
> (1) Binarization is the most aggressive quantization technique compared to multi-bit quantization. With 1-bit compact parameters in binary SR models, XNOR-POPCNT bitwise operations can be employed during inference, leading to up to 58x acceleration on ARM64 CPUs. In contrast, multi-bit quantized SR models can only utilize integer operations during inference.
>
> (2) Binarization can be seen as an orthogonal technique for distillation. On the one hand, the DBT distillation technique in FRB assists the student binarized network in learning information from the full-precision counterpart. On the other hand, distillation techniques can further aid in slimming SR networks from an architectural perspective, synergistically enhancing the efficiency of SR models alongside the extreme compression bit-width brought by binarization.
>
> > **Q2:** The authors claim that residual binarization reduces the binarization error, which is intuitive. However, are there any quantitative results that show this?
>
> **A2:** Our quantitative results demonstrate that FRB significantly improves accuracy by reducing binarization errors. As shown in Table 4, compared to vanilla binarization (DoReFa), SRB exhibits improvements in PSNR for B100, Urban100, and Manga109 datasets by 0.52, 1.41, and 0.85, respectively. Furthermore, Figure 1 in our supplementary material illustrates different representations of binarized weights, indicating that FRB notably enhances the binarized representations to resemble the weights in the full-precision counterparts closely.
>
> > **Q3:** What is the acceleration effect on actual hardware? The author only discusses the reduction in FLOPs of binary SR networks. I am curious about the acceleration on real hardware.
>
> **A3:** We followed the reviewer's suggestion to present the real inference time on ARM CPU devices based on the daBNN framework in Table A3. We provide the inference time on ARM64 CPU Raspberry Pi 4 Model B of different binarization methods with the same backbone SRResNet. FRBC achieves significant PSNR gains on Set5 (×2) with less than 0.02% increased inference time. The time of 15-RB FRBC-S is less than the existing methods while achieving higher accuracy.
>
> | Methods | FP32 | BNN | DoReFa | Bi-Real | IR-Net | BAM | BTM | ReActNet | FRBC | FRBC-S |
> | - | - | - | - | - | - | - | - | - | - | - |
> | Time (s) | 2.91 | 0.62 | 0.63 | 0.63 | 0.63 | 0.66 | 0.64 | 0.63 | 0.66 | 0.59 |
> | PSNR | 38.00 | 32.25 | 36.76 | 37.27 | 32.32 | 37.21 | 37.22 | 37.26 | 37.71 | 37.68 |
>
> Table A3: Quantitative comparison (x2). We use 15 RBs in our FRBC-S.

---

> > ### Comment · Reviewer_iE9s · 2023-11-22
> >
> > Thanks for the authors response and all my concerns have been well addressed. I have no further comments.

---

### Official Review · Reviewer_LkP1 · 2023-10-30

**Soundness:** 3 good
**Presentation:** 4 excellent
**Contribution:** 4 excellent
**Rating:** 8
**Confidence:** 4

**Summary:**

This paper proposes a flexible residual binarization (FRB) method for image SR to solve high-frequency information loss caused by binarization. FRB includes a second-order residual binarization (SRB) to counter the information loss caused by binarization and also includes Distillation-guided Binarization Training (DBT) to align the contents of different bit widths. The extensive experiments and comparisons show that FRBC and FRBT achieve superior performance both quantitatively and visually.

**Strengths:**

First, the flexible residual binarization proposed in this manuscript is effective and efficient. As the most extreme quantization method, binarization discretizes weights and activations to the greatest extent (1-bit), making it difficult for filters and features to extract details, that is, the high-frequency information mentioned in the article. The manuscript uses the second-order residual binarization of weights to restore the feature extraction abilities of the binarized SR network and maintain the acceleration brought by the XNOR-POPCNT instruction in inference. More importantly, compared with existing binarized SR methods, such as ABC-Net and HORQ, residual operations on weights do not bring an inference burden (compared to activation residuals).

Secondly, unlike some existing quantized SR methods, the proposed FRB is structurally universal, which allows the method to be widely used in various structural variants. The proposed binarization method acts at the operator level without affecting the overall architecture of the model. As for distillation, the full-precision counterpart of the binarized network is used, which has the same structure. This makes the proposed distillation flexible for various architectures.

Third, the accuracy and efficiency results of FRB are outstanding. Regarding accuracy, FRB achieves SOTA performance and surpasses existing binarized SR methods by a convincing margin. Regarding efficiency, since the unquantized head and tail account for most of the computational consumption in the binarized SR network, the burden caused by residual binarization is very small. Furthermore, the proposed method produces good visual effects in addition to quantitative results.

Finally, the manuscript is well-written and understandable, and the figures and formulas are well-presented.

**Weaknesses:**

Although the improvement of the proposed binarized SR method is significant, the loss caused cannot be ignored.

It is necessary to discuss the challenges of binarization in low-level fields. The manuscript proposes a well-designed binarization method but seems to lack analyses of its motivation, especially the key to binarizing models for low-level tasks compared to that for high-level tasks.

The manuscript did not discuss the feasibility of the proposed method in actual deployments, such as whether the proposed FRB can be well implemented using open-source deployment libraries (such as Larq [1], daBNN [2]) on ARM CPU hardware with good binarization support. Although few existing SR works discuss this point, practical deployment is crucial for binarization methods.
[1] Larq: An open-source library for training binarized neural networks
[2] dabnn: A super fast inference framework for binary neural networks on arm devices

For the proposed binarization-aware distillation method, the motivation for using blockwise distillation granularity is unclear (why not the more flexible layerwise). The manuscript needs to do more discussion and analysis on this.

**Questions:**

Considering the aforementioned weaknesses, I suggest that the author answer the following questions:

1) What is the key to binarizing models for low-level tasks compared to that for high-level tasks?

2) Can the proposed FRB be deployed on edge devices, like ARM CPU devices? If so, what about the performance?

3) Why not use the more flexible layerwise distillation?

---

> ### Author Response · Authors · 2023-11-16
> **Response to Reviewer LkP1**
>
> We thank you for your professional and insightful comments. Our response to your suggestion can be found below:
>
> > **Q1:** Although the improvement of the proposed binarized SR method is significant, the loss caused cannot be ignored.
>
> **A1:** Thank you for pointing that out. Despite the accuracy gap, our binary SR model (FRB) has significantly improved efficiency and accuracy. This paves a hopeful path for deploying SR models on edge devices. We believe that our work will assist in continuously advancing binarized SR networks toward higher accuracy to achieve practical application in the future.
>
> > **Q2:** It is necessary to discuss the challenges of binarization in low-level fields. The manuscript proposes a well-designed binarization method but seems to lack analyses of its motivation, especially the key to binarizing models for low-level tasks compared to that for high-level tasks. What is the key to binarizing models for low-level tasks compared to that for high-level tasks?
>
> **A2:** We clarify that the primary challenge of binarization in low-level tasks lies in the loss of high-frequency information due to highly discretized data, making it hard to handle detailed textures formed by dense pixels. As the binarizer operates on all intermediate features (activations) per-pixel basis, restoring local textures of interest becomes more challenging than the overall information focused on by high-level prediction tasks. Specifically, as mentioned in the third paragraph of our introduction: (1) Since activations (inputs) are binarized before calculation, a high degree of discretization leads to losing the high-frequency information of the feature map. (2) Since the weights are binarized, the filter with reduced representation capability makes extracting high-frequency information from the feature map hard.
>
> > **Q3:** The manuscript did not discuss the feasibility of the proposed method in actual deployments, such as whether the proposed FRB can be well implemented using open-source deployment libraries (such as Larq [1], daBNN [2]) on ARM CPU hardware with good binarization support. Although few existing SR works discuss this point, practical deployment is crucial for binarization methods. Can the proposed FRB be deployed on edge devices, like ARM CPU devices? If so, what about the performance?
> [1] Larq: An open-source library for training binarized neural networks
> [2] dabnn: A super fast inference framework for binary neural networks on arm devices.
>
> **A3:** We followed the reviewer's suggestion to present the real inference time on ARM CPU devices based on the daBNN framework in Table A3. We provide the inference time on ARM64 CPU Raspberry Pi 4 Model B of different binarization methods with the same backbone SRResNet. FRBC achieves significant PSNR gains on Set5 (×2) with less than 0.02% increased inference time. The time of 15-RB FRBC-S is less than the existing methods while achieving higher accuracy.
>
> | Methods | FP32 | BNN | DoReFa | Bi-Real | IR-Net | BAM | BTM | ReActNet | FRBC | FRBC-S |
> | - | - | - | - | - | - | - | - | - | - | - |
> | Time (s) | 2.91 | 0.62 | 0.63 | 0.63 | 0.63 | 0.66 | 0.64 | 0.63 | 0.66 | 0.59 |
> | PSNR | 38.00 | 32.25 | 36.76 | 37.27 | 32.32 | 37.21 | 37.22 | 37.26 | 37.71 | 37.68 |
>
> Table A3: Quantitative comparison (x2). We use 15 RBs in our FRBC-S.
>
> > **Q4:** For the proposed binarization-aware distillation method, the motivation for using blockwise distillation granularity is unclear (why not the more flexible layerwise). The manuscript needs to do more discussion and analysis on this. Why not use the more flexible layerwise distillation?
>
> **A4:** Despite the nearly identical architecture between full-precision and binarized models, the substantial gap in capabilities at each layer between the two may render it challenging for a student, during layer-wise distillation, to mimic the teacher's hidden representations [ref1]. Furthermore, the hidden representations of a full-precision teacher typically contain a considerable amount of redundant information [ref2]. Due to the limited capacity of the binarized student, this redundancy might compete with useful information, impeding the extraction of valuable knowledge.
>
> [ref1] Less is More: Task-aware Layer-wise Distillation for Language Model Compression. ICML2023.
>
> [ref2] Analyzing redundancy in pretrained transformer models. EMNLP 2020.

---

> ### Comment · Reviewer_LkP1 · 2023-11-22
> **feedback**
>
> Thanks for your responses. My concerns have been well solved, and I have no further questions. Thus, I tend to vote for accepting it.

---

### Official Review · Reviewer_Bydn · 2023-11-01

**Soundness:** 3 good
**Presentation:** 2 fair
**Contribution:** 2 fair
**Rating:** 8
**Confidence:** 4

**Summary:**

This paper proposes an efficient super-resolution method with a binarization quantization technology to address insufficient high-frequency information and distortion of representation content, namely flexible residual binarization (FRB). The FRB designs two components to build the total model, including a Second-order Residual Binarization (SRB) for countering the information loss caused by binarization and the Distillation-guided Binarization Training (DBT) for narrowing the representation content gap between the binarized and full-precision networks. Furthermore, the author generalizes the proposed FRB model by applying the model to binarize convolution and Transformer-based SR networks. The author declares that they conduct extensive experiments on benchmark datasets to prove the effectiveness of the proposed FRB.

**Strengths:**

1. This paper introduces a pretty nice quantization technology to achieve efficient super-resolution tasks by a binarization quantization strategy.
2. The motivation of this paper is clearly exhibited in the abstract section, which contains two parts, (1) insufficient high-frequency information and (2) representation content gap between the binarized and full-precision networks.
3. The proposed FRB is applied to the binarized convolution and Transformer-based SR networks, which is an admirable expression.

**Weaknesses:**

1. The two parts describe the same issue which is the loss of high-frequency information caused by the binarization operation in the third paragraph of the introduction, which is different from the motivations described in the abstraction section.
2. In the DBT module, the dense middle feature distillation is a widely used technology in many tasks. The author didn’t report the difference with general knowledge distillation.
3. The author should briefly introduce the used datasets, including the size and annotations of datasets.
4. The quantitative is not fair. The adopted pipelines (SRResNet and SwinIR_S) were also binarized without proposed work (SRD and DBT) in this paper, which can be a counterpart in the same setting.
5. The parameters and FLOPs of compared methods are absent in Table 1. Besides, the FPS metric should be considered.
6. Ablation studies are not sufficient. For example, to prove the effectiveness of SRB, the author needs to remove the residual binarization, and then compare it with the complete model. Moreover, SRB can be viewed as a plug-and-play module, which embeds into other methods to prove its effectiveness.
7. The references lack new literature published in 2022 and 2023.

**Questions:**

1. In the SRB module, values of the reconstruction error (residual binarize) surpass the binarization. In general, the weight binarization undertakes the main function, the additional constraint (or information) is secondary. But how to explain the effect of this situation?
2. Which GPU was used to train the proposed model in the training Strategy?

---

> ### Author Response · Authors · 2023-11-16
> **Response to Reviewer Bydn (1/2)**
>
> We thank the reviewer for the feedback and comments. We respond to the concerns below:
>
> > **Q1:** The two parts describe the same issue which is the loss of high-frequency information caused by the binarization operation in the third paragraph of the introduction, which is different from the motivations described in the abstraction section.
>
> **A1:** We would like to clarify that, as our abstract states, binarization induces high-frequency information loss, and the two parts of the third paragraph of the introduction explain why.
>
> For SR networks in which both weights and activations are binarized, high-frequency losses exist in two aspects: (1) Since activations (inputs) are binarized before calculation, a high degree of discretization leads to losing the high-frequency information of the feature map. (2) Since the weights are binarized, the filter with reduced representation capability makes extracting high-frequency information from the feature map hard.
>
> The loss of high-frequency information from weights and activations caused by binarization to the SR network brings direct motivation to our proposed FRB method, that is, the ability to extract high-frequency information by restoring the weights through SRB and restore the high-frequency information from the activation through DBT information.
>
> > **Q2:** In the DBT module, the dense middle feature distillation is a widely used technology in many tasks. The author didn’t report the difference with general knowledge distillation.
>
> **A2:** Our DBT is a direct technique to improve the problem of high-frequency information loss in activation. It restores the high-frequency information expression capability of the binary SR model through the feature map of the full-precision counterpart. Since the binarization operation is performed at each layer, we use a more dense block-wise distillation form and a normalized attention form to stabilize the contents in networks of different bit-widths uniformly. Our experiments prove that our proposed DBT significantly improves the performance of binary SR models.
>
> > **Q3:** The author should briefly introduce the used datasets, including the size and annotations of datasets.
>
> **A3:** We further supplemented the detailed information of the dataset according to the reviewer's suggestions.
>
> Following the common practice in image SR, we adopt DIV2K as the training data, which contains 800 training images, 100 validating images, and 100 testing images. Five benchmark datasets are used for testing: Set5, Set14, B100, Urban100, and Manga109. Specifically, the Set5 dataset consists of 5 images (“baby”, “bird”, “butterfly”, “head”, “woman”). The Set14 dataset consists of 14 images. B100 represents the set of 100 testing images from the Berkeley Segmentation Dataset [ref1]. The  Urban100 dataset contains 100 images of urban scenes. The Manga109 dataset is composed of 109 manga volumes drawn by professional manga artists in Japan.
>
> [ref1] D. Martin, et al. A database of human segmented natural images and its application to evaluating segmentation algorithms and measuring ecological statistics. ICCV 2001.
>
> > **Q4:** The quantitative is not fair. The adopted pipelines (SRResNet and SwinIR_S) were also binarized without proposed work (SRD and DBT) in this paper, which can be a counterpart in the same setting.
>
> **A4:** We clarify that our comparisons are based on the exact same architecture (Table 1: SRResNet, Table 2: SwinIR_S) to maintain a fair comparison of different binary quantization methods, and the other settings are exactly the same. Taking Table 1 as an example, FRBC and FRBC+ are SRResNets that use the binarization of our proposed FRB (including SRB and DBT technologies). Other methods with a bit width of 1/1 use existing binarization methods. , the 32/32-bit wide SRResNet corresponds to a full-precision copy (original model). Those in the same table have exactly the same architecture and training settings, so the comparison is fair.

---

> ### Author Response · Authors · 2023-11-16
> **Response to Reviewer Bydn (2/2)**
>
> > **Q5:** The parameters and FLOPs of compared methods are absent in Table 1. Besides, the FPS metric should be considered.
>
> **A5:** We followed the reviewer's suggestion to compare the Ops and Params of binarization methods and their inference time on real ARM CPU devices.
>
> In Tab. A5, we compare the Ops and Params of our FRBC with the other methods. Our FRB achieves significant improvements with little overhead increases. We retrain our FRBC with scale x2 to keep much fewer parameters than other methods by removing one residual block (RB) in SRResNet, namely 15 RBs. We denote this version as FRBC-S. The results of FRBC-S still significantly outperform other methods.
>
> Moreover, we also provide the inference time on ARM64 CPU Raspberry Pi 4 Model B of different binarization methods with the same backbone SRResNet. FRBC achieves significant PSNR gains on Set5 (×2) with less than 0.02% increased inference time. The time of 15-RB FRBC-S is less than the existing methods while achieving higher accuracy.
>
> | Methods | FP32 | BNN | DoReFa | Bi-Real | IR-Net | BAM | BTM | ReActNet | FRBC | FRBC-S |
> | - | - | - | - | - | - | - | - | - | - | - |
> | Time (s) | 2.91 | 0.62 | 0.63 | 0.63 | 0.63 | 0.66 | 0.64 | 0.63 | 0.66 | 0.59 |
> | Ops (G)     | 85.4 | 18.6 | 18.6 | 18.6 | 18.6 | 18.6 | 18.6 | 18.6 | 19.7 | 17.5 |
> | Params (K) | 1367 | 225 | 225 | 225 | 225 | 225 | 225 | 225 | 238 | 223 |
> | PSNR | 38.00 | 32.25 | 36.76 | 37.27 | 32.32 | 37.21 | 37.22 | 37.26 | 37.71 | 37.68 |
>
> Table A5: Quantitative comparison (x2). We use 15 RBs in our FRBC-S. Input size is 3x320x180 for Ops calculation.
>
> > **Q6:** Ablation studies are not sufficient. For example, to prove the effectiveness of SRB, the author needs to remove the residual binarization, and then compare it with the complete model. Moreover, SRB can be viewed as a plug-and-play module, which embeds into other methods to prove its effectiveness.
>
> **A6:** We would like to clarify that since SRB is a technology that acts on the binarization operator, this technology is completely parallel to other binarization algorithms. Specifically, when the residual binarization is removed, FRB degenerates into quantization in the form of DoReFa (i.e., the first row of results in Table 4). When it is replaced with other binarizations, it degrades to the results of other models in Table 1.
>
> > **Q7:** The references lack new literature published in 2022 and 2023.
>
> **A7:** In Table 1, we compare the SOTA binarization-aware training method BBCU [ref2], which is published in ICLR 2023, with our FRB, and the results show that our FRB outperforms it in various scales and datasets.
>
> As suggested, we will also include more recent works (like [ref3]) in the paper.
>
> [ref2] Basic binary convolution unit for binarized image restoration network. ICLR 2023.
>
> [ref3] Toward Accurate Post-Training Quantization for Image Super Resolution, CVPR 2023.
>
> > **Q8:** In the SRB module, values of the reconstruction error (residual binarize) surpass the binarization. In general, the weight binarization undertakes the main function, the additional constraint (or information) is secondary. But how to explain the effect of this situation?
>
> **A8:** We clarify that our original manuscript does not state the relative relationship between the reconstruction error and the original binarized value. The values of the original binarization (Bw1 in Eq. 7) and reconstruction error (Bw2 in Eq. 8) in SRB are based on statistics, aiming to use the second-order residual to more closely approximate the real-value weight (compared to the direct binary value). And since scaling factors are used to minimize the binarization error at each order of approximation (α1 and α1 for Bw1 and Bw2, respectively), the numerical sizes of Bw1 and Bw2 are directly related to the original weight w and the residual ||w-Bw1||.
>
> > **Q9:** Which GPU was used to train the proposed model in the training Strategy?
>
> **A9:** We use a single NVIDIA A6000 GPU to train each model.

---

### Comment · Reviewer_Bydn · 2023-11-23
**feedback**

Thanks for the detailed explanation from the authors. Since my concerns have been well solved, I decide to raise my score and suggest to accept this work.

---

### Meta-Review · Program_Chairs · 2023-12-09

**Metareview:**

This meta review is written by the PCs.

This paper presents a “flexible” residual binarization for image Super resolution. The main problem is stated to be the loss of “high-frequency” information due to binarization of weights and activations both. The method is pretty straightforward: add a binarization of the residual error. Standard techniques of distillation is used during training.

The novelty is pretty low. Adding more information (from residual error) of course should increase the performance but it also increases the overhead. One can balance this to keep the overhead low, but the originality of idea is low. It is not clear why the method is “flexible”: it is a heavy word used without a clear explanation. Also the use of word “second-order residual binarization” is also unnecessary: the proposal is to simply add additional information.

It is troubling that the reviews are not critical or informative. The reviewers high score is not backed by any arguments that shows that the idea is original and the paper is a strong one, worthy of publishing at ICLR. The reviewers’ feedback is also simply one line and there is a lack of discussion. The AC report is also mildly supportive of the paper, again, without any strong reasoning.

PCs discussed the paper among themselves and, due to the reasons outlined above, did not find any strong compelling reason to accept the paper.

**Justification For Why Not Higher Score:**

The binarization technique itself is useful, however, but the validated task is limited to single-image super-resolution. The contributions could have been made stronger if the gains were validated in more general low-level vision tasks.

**Justification For Why Not Lower Score:**

Reviewers recognized the value of this work and admitted the gains from the residual binarization which is quite novel compared with the previous network binarization techniques.

---

### Decision · Program_Chairs · 2024-01-16

Reject